# Non-invasive assessment of steatohepatitis indicates increased risk of coronary artery disease

Sebastian Beer[1,2☯], Jonas Babel[1,3☯], Neef Martin[4], Valentin Blank[1,5], Johannes Wiegand[6], Thomas Karlas[1]*

1 Department of Medicine II, Division of Gastroenterology, Leipzig University Medical Center, Leipzig, Germany, 2 Integrated Research and Treatment Center Adiposity Diseases, Faculty of Medicine, University of Leipzig, Leipzig, Germany, 3 Department of Operative Medicine II, Division of Visceral-, Transplant-, Thoracic- and Vascular Surgery, Leipzig University Medical Center, Leipzig, Germany, 4 Department of Cardiology, Leipzig University Medical Center, Leipzig, Germany, 5 Department of Internal Medicine I (Gastroenterology, Pneumology) and Division of Interdisciplinary Ultrasound, University Hospital Halle (Saale), Halle (Saale), Germany, 6 Department of Medicine II, Division of Hepatology, Leipzig University Medical Center, Leipzig, Germany

☯ These authors contributed equally to this work.
* Thomas.Karlas@medizin.uni-leipzig.de

**Data Availability Statement:** All relevant data are within the manuscript and its Supporting Information files (S1 Dataset).

## Abstract

### Introduction

Fatty liver diseases (FLD), especially defined as metabolic dysfunction-associated FLD (MAFLD), is of growing importance for patients and health-care providers. Extrahepatic comorbidities, predominantly coronary artery disease (CAD), contribute to excess morbidity and mortality in FLD. Although the association of FLD and CAD is well known, underlying pathophysiological links are not fully understood. Non-invasive means of liver diagnostic enable a fast and thorough characterization of FLD. We therefore assessed the severity of FLD in a cohort of patients at risk of CAD.

### Methods

Patients scheduled for coronary angiography were characterized by anthropometry, serum-based indices of liver fibrosis (NFS, FIB4), abdominal ultrasound and vibration controlled transient elastography (VCTE) including controlled attenuation parameter (CAP) and the Fibroscan-AST (FAST) score. Patients were stratified according to indication of therapeutic coronary intervention.

### Results

120 patients were recruited, MAFLD was found in 41%, while advanced fibrosis or cirrhosis were present in only 5%. Coronary vascular intervention was indicated in 42% (n = 50). Severity of steatosis assessed by CAP and risk of fibrosis defined by elevated liver stiffness (VCTE>8 kPa) and fibrosis indices were associated with the need for coronary intervention. FAST score, a marker of fibrotic steatohepatitis, was elevated in the intervention group

**Funding:** This work was supported by the Federal Ministry of Education and Research (BMBF) by a MetaRot Grant of the Integrated Research and Treatment Center AdiposityDiseases Leipzig (IFB), Germany, FKZ01EO1001 to SB. https://www.ifb-adipositas.de/en The funders had no role in study design, data collection and analysis, decision to publish, or preparation of the manuscript.

**Competing interests:** I have read the journal's policy and the authors of this manuscript have the following competing interests: TK and JW received an unrestricted research grant from Echosens, France. TK was an invited speaker for Echosens. SB, JB, MN and VB declared that no competing interests exist. This does not alter our adherence to PLOS ONE policies on sharing data and materials.

(0.22 vs. 0.12, p<0.001). Multivariate regression analysis revealed FAST score as strongest predictor of CAD (OR 2.3 95%, CI 1.40–2.96).

## Discussion

MAFLD is a frequent comorbidity in patients at CAD risk, but advanced liver disease has a low prevalence in patients undergoing elective coronary angiography. Therefore, a routine VCTE-based screening for FLD cannot be recommended in cardiac patients. The association of indicators of steatohepatitis with advanced CAD points to inflammatory processes as a conjoint mechanism of both diseases.

## Introduction

The metabolic syndrome (MS) is a composition of cardiovascular risk factors–namely adiposity, high blood pressure, impaired glucose and fat metabolism–and represents a major burden to both the individual patient and society [1]. Organ-specific complications comprise type 2 diabetes, coronary artery disease (CAD), obstructive sleep apnea or fatty liver disease (FLD), among others [1, 2]. Of these, cardiovascular disease (CVD) is the leading cause of death in Europe, therefore prevention and early treatment are crucial [3].

Another important manifestation of the MS is fatty liver disease (FLD), i.e. non-alcoholic fatty liver disease (NAFLD) [1, 4, 5] and accordingly metabolic associated fatty liver disease (MAFLD) [6]. The currently introduced MAFLD definition puts emphasis on metabolic disorders resulting in FLD, regardless of alcohol consumption [6].

Complications of FLD comprise development of steatohepatitis, which can progress to fibrosis and cirrhosis [7]. Moreover, patients with FLD have an increased risk to develop hepatocellular carcinoma, even in cases without advanced fibrosis or cirrhosis [8–10]. Therefore, patients with advanced FLD show a higher liver specific morbidity and mortality [11]. In addition, NAFLD patients also show a higher overall mortality due to extrahepatic reasons, e.g. CAD [11, 12].

Several studies suggest an association between severity of FLD and presence of relevant CAD, which may be induced by similar risk factors [12–14]. A recent meta-analysis revealed a prevalence of 44.6% (95% confidence interval (CI), 36.0%–53.6%) of CAD among FLD-patients [15]. Interestingly, the prevalence of CAD increased to 60.5% when invasive angiography was used (95% CI, 43.8%-75.1%). There is strong evidence that NAFLD is an independent risk factor for cardiovascular events [16]. Furthermore, some studies suggest an association between the severity of NAFLD and CAD risk [12, 15–17].

In clinical practice, prevention, early detection, and treatment of FLD and CAD are important but specific preventive measures are not always in awareness or available [18–20]. Moreover, risky alcohol consumption is frequent among patients with CAD and there is evidence for an increased cardiovascular risk among heavy drinkers [21]. Such patients are neglected by risk stratification algorithms of traditional NAFLD guidelines [20]. Therefore, the MAFLD definition better reflects real life scenarios than the traditional NAFLD definition and is thus increasingly applied [22].

A major problem for studying the relationship between FLD and CAD is the need for a reliable liver tissue characterization because grade of steatosis, inflammatory activity and grade of fibrosis provide important information on disease severity and risk of progression. The traditional gold standard liver biopsy is, however, associated with bleeding risk, especially in CAD

patients due to antiplatelet or anticoagulation drugs and, therefore, restricted to carefully selected patients [23]. Recent developments of non-invasive ultrasound-based methods now allow a thorough FLD screening in CAD patients at a larger scale [24]. Elastography [25] combined with attenuation quantification [26] and risk score calculation [27] provide reliable and fast tools for estimation of fibrosis severity and steatosis grade as well as risk assessment of steatohepatitis.

In this pilot study, we non-invasively investigated the presence and especially the severity of FLD, patients at MAFLD risk, in a cohort with suspicion of advanced CAD.

## Patients and methods

### Study design and patient selection

This prospective, cross-sectional, single-center, diagnostic study was designed to evaluate the interplay between FLD and CAD risk.

The study protocol was approved by the ethical committee of the University Leipzig (registration number 148-15-09032015) and registered in the Clinical Trials Register of the U.S. National Library of Medicine (ClinicalTrials.gov, NCT02779946). All patients provided written informed consent.

Between October 2015 and February 2017 adult patients (≥18 years) with indication for coronary angiography (CA) were recruited at the Division of Cardiology of Leipzig University Medical Center. We considered patients either scheduled for routine angiography or undergoing emergency angiography due to (suspected) myocardial infarction (MI). In the latter case, patients were only considered after successful cardiological therapy with stabilized heart function prior to discharge from hospital.

Patients who have had known CA before were excluded from the analysis. Further exclusion criteria comprised age < 18 years, pregnancy or lactation, active malignant disease within 12 months before inclusion, patients with a liver transplant or altered liver anatomy, e.g. after resection of right liver lobe, cholestasis on ultrasound imaging, severe congestive heart failure (EF <30%, NYHA III or IV, diastolic dysfunction III or IV) and pulmonary hypertension (WHO III or IV) [28, 29]. In addition, patients with history of chronic viral hepatitis were not considered.

### Study examination

The study examinations were performed immediately prior to or within 21 days after the cardiological intervention but did not delay any cardiological diagnostics. Ultrasound and liver risk assessment were performed by a blinded examiner (JB and SB) to the results of CA to avoid possible investigator bias.

All patients underwent a thorough clinical characterization including case history, anthropometric data, lifestyle, alcohol and nicotine consumption and medication. Alcohol consumption was recorded as average alcohol intake per week (grams). From this, the daily alcohol consumption was derived. For sub-analysis, patients were stratified according to risky drinking behavior. Average alcohol intake of >20g/day in men and >10g/day in women was considered relevant according to the German NAFLD guideline [20]. For the laboratory data, a recent blood sample in an overnight fasting state with a maximum interval of 3 weeks was required.

Additionally, clinically established risk scores (NAFLD-fibrosis score (NFS), FIB-4-index (FIB4) as well as the newly developed FibroScan-AST score (FAST), *see below*) were calculated. Prior to CA–or in case of emergency CA after clinical stabilization within 21 days–, a thorough abdominal ultrasound examination including liver stiffness measurement (LSM) with

vibration controlled transient elastography (VCTE) combined with measurement of controlled attenuation parameter (CAP) were performed by an experienced examiner using Siemens Acuson S2000 ultrasound device (linear and curved array probe). A fasting period for at least four hours prior to examination was required.

MAFLD was defined by the presence of elevated CAP (see below) and at least one of either overweight/obesity (BMI >25kg/m$^2$) or diabetes mellitus type 2 or the presents of at least two metabolic risk abnormalities [30].

## Vibration controlled transient elastography

VCTE (kPa) of the liver including quantification of steatosis with CAP (dB/m) was performed by a trained examiner as previously described [31]. In brief, ten valid measurements were performed in supine position in a right intercostal space according to the manufacturer's recommendation [25]. LSM were performed using the M- and XL-probe of the Fibroscan® (FibroScan; Echosens, Paris, France) according to the skin-liver-capsule distance (SLCD) [32]. The SLCD was measured by a conventional linear ultrasound transducer. Patients with SLCD exceeding 25mm were examined with the XL-probe. Focal lesions affecting the measurement area of VCTE, hepatic cholestasis, presence of ascites as well as liver congestion due to right heart failure were ruled out by conventional ultrasound (see above) [25].

An interquartile range (IQR) of less than 30% in patients with LSM >7.1 kPa was required [33]. According to the German NAFLD guideline, LSM values of ≥8 kPa defined an increased risk for significant fibrosis [20, 24]. CAP values ≥302 dB/m indicated presence of steatoses (any grade), a value ≥ 331 dB/m indicated advanced steatosis (S2/3) [34].

## Laboratory values and clinical risk scores

We recorded full blood count, alanine and aspartate aminotransferases (ALT and AST), gamma-glutamyl-transferase (GGT), alkaline phosphatase (AP), albumin, creatinine, calculated glomerular filtration rate (GFR), lipid profile (cholesterol, high- and low-density lipoprotein (LDL and HDL) triglycerides) and glycohemoglobin (HbA1c). The upper limit of normal of AST at time of the study was 0.6 µkat/l in women and 0.85 µkat/l in men, and of ALT 0.58 µkat/l in women and 0.85 µkat/l in men, respectively.

Clinically well-established fibrosis risk scores (NFS; FIB4) were calculated [35, 36]. For NFS, sensitive/specific cut-offs were −1.455 (age 36–65) and 0.12 (age ≥ 65)/0.676 (age ≥ 36). For FIB4, sensitive/specific cut-offs were 1.3 (age < 65) and 2.0 (age ≥ 65)/2.67 (all ages), respectively.

FAST is a recently developed elastography based tool for assessing the risk of active fibrotic (≥F2) steatohepatitis (NAS≥4) [27]. We calculated FAST according to the published formula using CAP, LSM and AST. We applied sensitive (0.35) and specific (0.67) cutoff values for the detection and exclusion of active fibrotic steatohepatitis [27].

## Cardiological diagnostics and coronary angiography

Results of routine echocardiography were used to rule out right heart stress and severe insufficiency. CA was performed by an experienced interventionalist according to current recommendations [37] using the radial or femoral artery as vascular access. Severity of CAD was evaluated according to the grade of stenosis (wall irregularities, moderate stenosis of ≥ 50% lumen reduction and severe stenosis of ≥ 75% lumen reduction, respectively) [37].

The indication for revascularization was based on the angiography findings as well as the patient's clinical presentation in accordance with the valid guideline at the time of intervention [38, 39].

For the main data analysis, patients were divided into two groups based on the need for revascularization: CAD requiring revascularization (percutaneous transluminal coronary angioplasty (PTCA), stenting or cardiac bypass) and no intervention needed.

## Statistics

Statistical analyses were performed using SPSS Version 27. Values are displayed as median and interquartile range (IQR) by X (X, Y) and absolute numbers and percentage by X (Y%). Chi-square test without continuity correction was used to evaluate association between two categorical variables. Mann-Whitney U-test was used to compare medians between two non-normally distributed groups. Logistic regression was used to identify independent risk factors for coronary intervention using the software R. For logistic regression, a p-value was based on the Wald statistic and the 95% CI determined with a profiling method.

A p-value of <0.05 was considered significant.

## Results

A total of 216 patients referred for CA were screened. After exclusion of 96 patients due to various reasons, 120 patients were available for the final analysis (Fig 1). Of these 120 patients, n = 19 (15.8%) underwent emergency CA due to (suspected) myocardial infarction. Indication for routine CA (n = 101) was stable angina pectoris.

Baseline characteristics of the total study cohort and the MAFLD sub-cohort are shown in Table 1, stratified for the presence or absence of MAFLD. Using only CAP for steatosis detection 49 patients (41%) were diagnosed with MAFLD as defined by the expert consensus statement [37]. A total of 50 patients (42%) of the total cohort required intervention including nine patients being referred for coronary bypass surgery. The remaining 70 patients had no indication for coronary intervention. In the MALFD subgroup the proportion of patients requiring intervention was comparable with the non MAFLD subgroup: 45% (22 patients) vs. 39% (28 patients; p = 0.55).

### Assessment of fatty liver disease by elastography

LSM including CAP could not be applied in four patients due to cysts of the right liver lobe (n = 1) or insufficient fasting status (n = 3). Cases with invalid (less than 10 single measurements; n = 1) or unreliable (n = 7) LSM had high BMI (median of 34 kg/m$^2$ vs. 28 kg/m$^2$; p = 0.002) or live-capsule distance (median of 27 mm vs. 21 mm; p = 0.002). Thus, LSM was valid and reliable in 120 (94% of eligible) patients (Fig 1).

Table 2 shows LSM and CAP results stratified by the results of coronary diagnostics. Median LSM values were within the normal range but differed significantly between the subgroups (intervention vs. no intervention: 4.9 kPa vs 4.2 kPa; p = 0.018). Suspicion of significant fibrosis in the overall cohort was found rarely in only 6 cases (5%), also equally distributed between the groups. Results in the MAFLD subgroup were similar (S1 Table).

Steatosis detected by CAP could be seen in 50 patients (41.7%) of the total cohort with 28 patients (23.3%) with signs of moderate or severe steatosis (≥S2). CAP values did not differ significantly between the groups (see Table 2 and S1 Table).

### Laboratory based risk indices

Calculation of the NFS revealed a relatively high proportion of patients above the sensitive cut-off (Table 2). No difference between the groups was seen (31%, n = 15 in the interventional group vs. 29%, n = 20; p = 0.79). Only 5 patients (10%) of the interventional group and 4

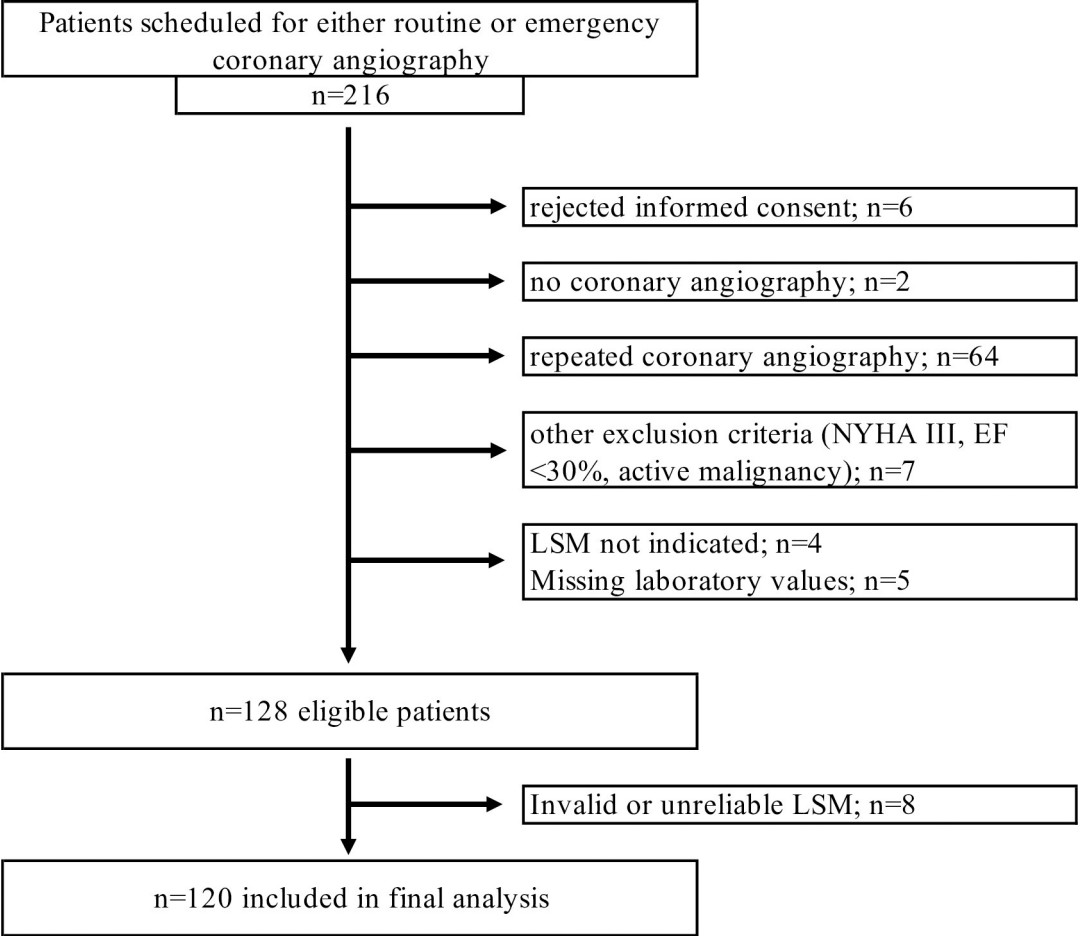

**Fig 1. Summary of patient recruitment.** *NYHA* New York Heart Association, *EF* Ejection fraction; *LSM* Liver stiffness measurement.

patients (6%) of the non-intervention group (p = 0.36) were identified as high-risk according to the specific cut-off of NFS. Similar numbers were seen in the MALFD group (S1 Table).

FIB4 did neither show a significant difference between the groups using the sensitive (51%, n = 25 vs. 41%, n = 28; p = 0.26) nor the specific cut-off (14%, n = 7 vs. 4%, n = 3, p = 0.056).

Unexpectedly, there was a poor agreement in risk assessment between LSM, FIB4 and NFS: none of the high-risk FIB4 patients and only one of the high-risk NFS patients had increased LSM values. The small number of patients with significant elevated LSM ≥8kPa had significantly higher BMI, waist-to-hip ratio and CAP resulting in an increased prevalence of MAFLD and elevated FAST score (S2 Table).

### Risk of fibrotic steatohepatitis

The non-invasive evaluation of fibrotic steatohepatitis using the FAST score revealed significantly higher (median 0.22 vs. 0.12; p < 0.001) values in patients with indication for coronary intervention (Table 2 and S1 Table). This difference persisted after exclusion of n = 11 patients with myocardial-infarction-to-LSM-interval below one week (n = 109; median 0.22 vs. 0.12; p = 0.012). In addition, we performed a multivariate logistic regression analysis of the whole patient sample taking known anthropometric and behavioral parameters including alcohol

**Table 1. Baseline characteristics of the total cohort stratified for the presence or absence of MAFLD.**

| Variables | Total cohort | | non-MAFLD group | | MAFLD group | | p-value |
|---|---|---|---|---|---|---|---|
| | n = 120 | | n = 71 (59%) | | n = 49 (41%) | | |
| Age, years | 65.2 | (58.1;73.6) | 64.0 | (55.7;73.2) | 66.4 | (58.9;74.5) | 0.47 |
| Male gender | 80 | (66.7%) | 42 | (59.2%) | 38 | (77.6%) | **0.036** |
| BMI, kg/m² | 27.6 | (25.0;30.4) | 25.7 | (24.0;28.1) | 30.5 | (28.4;34.6) | **<0.001** |
| BMI >25 kg/m² | 90 | (75.0%) | 43 | (60.6%) | 47 | (95.9%) | **<0.001** |
| Relevant alcohol consumption | 50 | (41.7%) | 30 | (42.3%) | 20 | (40.8%) | 0.88 |
| High blood pressure | 86 | (71.7%) | 49 | (69.0%) | 37 | (75.5%) | 0.44 |
| Diabetes mellitus | 29 | (24.2%) | 11 | 1(5.5%) | 19 | (38.8% | **0.004** |
| Waist-hip ratio | 0.97 | (0.91;1.02) | 0.95 | (0.89;1.00) | 1.00 | (0.96;1.03) | **0.001** |
| CAD | 50 | (41.7%) | 28 | (39.4%) | 22 | (44.9%) | 0.55 |
| hs-CRP, mg/l | 2.79 | (1.33;7.23) | 2.11 | (0.82;6.43) | 3.38 | (1.85;7.27) | 0.18 |
| Trigylcerides, mmol/l | 1.34 | (0.98;1.78) | 1.11 | (0.91;1.54) | 1.70 | (1.29;2.84) | **<0.001** |
| HDL cholesterol, mmol/l | 1.38 | (1.11;1.71) | 1.48 | (1.20;1.83) | 1.24 | (1.02;1.39) | **<0.001** |
| LDL cholesterol, mmol/l | 3.30 | (3.68;4.30) | 3.30 | (2.65;4.06) | 3.36 | (2.84;4.49) | 0.21 |
| AST, µkat/l | 0.45 | (0.39;0.57) | 0.46 | (0.39;0.58) | 0.45 | (0.40;0.57) | 0.76 |
| AST % of ULN | 60.0 | (50.6;76.7) | 63.3 | (50.6;86.0) | 58.8 | (50.6;68.3) | 0.24 |
| AST > ULN (n) | 12 | (10%) | 9 | (12.7%) | 3 | (6.1%) | 0.24 |
| ALT, µkat/l | 0.41 | (0.33;0.57) | 0.40 | (0.31;0.57) | 0.44 | (0.35;0.62) | 0.21 |
| ALT % of ULN | 57.3 | (45.6;75.9) | 56.9 | (43.5;75.0) | 58.8 | (48.2;77.6) | 0.83 |
| ALT > ULN (n) | 11 | (9,2%) | 6 | (8,5%) | 6 | (12,2%) | 0.50 |
| GGT, µkat/l | 0.50 | (0.35;0.73) | 0.49 | (0.32;0.71) | 0.57 | (0.41;0.73) | 0.19 |
| AP, µkat/l | 1.14 | (0.97;1.37) | 1.14 | (0.99;1.34) | 1.14 | (0.95;1.41) | 0.90 |
| INR | 1.0 | (1.0;1.1) | 1.0 | (1.0;1.1) | 1.0 | (0.9;1.1) | 0.97 |
| PTT, sec | 29.5 | (28.0;31.5) | 29.1 | (27.8;30.6) | 29.8 | (28.6;32.6) | **0.044** |
| Albumin, g/l | 45.1 | (43.4;46.7) | 45.1 | (43.2;47.2) | 44.9 | (43.6;46.5) | 0.62 |
| Platelets, 10⁹/l | 231 | (201;284) | 232 | (200;289) | 229 | (205;277) | 0.91 |
| Bilirubin, µmol/l | 8.5 | (6.5;11.5) | 8.4 | (6.4;11.5) | 8.6 | (6.9;11.9) | 0.96 |
| HbA1c, % | 5.56 | (5.36;5.98) | 5.54 | (5.31;5.74) | 5.61 | (5.46;6.39) | **0.011** |

Values given in median (IQR) and absolute numbers (%)

MAFLD, metabolic associated fatty liver disease; BMI, Body Mass index; CAD, coronary artery disease; hs-CRP, high-sensitive C-reactive protein; HDL cholesterol, high-density lipoprotein cholesterol; AST, aspartate aminotransferase; ALT, alanine transaminase; ULN, upper limit of normal; gGT, gamma-glutamyltransferase; AP, alcalic phosphatase; INR, international normalized ratio of prothrombin time; PTT, partial thromboplastin time; HbA1c, glycated haemoglobin

consumption into account (Table 3). In the MAFLD subgroup, the values were similar although the limited patient number did not allow a multivariate logistic regression.

## Discussion

FLD and CAD are both associated with the metabolic syndrome, but their pathophysiological interplay is not fully clarified [40]. Prevalence of FLD is increasing, especially since the MAFLD incorporated many patients with coexisting other liver disease [41]. FLD is associated with increased liver related morbidity and mortality in patients with advanced liver disease, whereas cardiovascular events are a leading cause of mortality in FLD patients without relevant fibrosis [42]. Therefore, patients with FLD should be carefully assessed for co-existing CAD [18, 20]. However, there is less evidence available if patients at high risk for CAD require intensified FLD screening.

**Table 2. Severity of liver disease.**

| Variables | | Total cohort | | Coronary Intervention | | | | | |
|---|---|---|---|---|---|---|---|---|---|
| | | | | Yes | | No | | | |
| | | n = 120 | | n = 50 | | n = 70 | | p-value | |
| **Steatosis risk** | | | | | | | | | |
| CAP (dB/m) | | 289 | (245;327) | 290 | (263;340) | 289 | (244;318) | 0.43 | |
| S0 | CAP ≤ 302 dB/m | 70 | (58.3%) | 28 | (56%) | 42 | (60%) | 0.66 | |
| S1 | 302 < CAP < 331 dB/m | 22 | (18.3%) | 8 | (16%) | 14 | (20%) | 0.58 | |
| S2/3 | CAP ≥ 331 dB/m | 28 | (23.3%) | 14 | (28%) | 14 | (20%) | 0.31 | |
| **Fibrosis risk** | | | | | | | | | |
| Liver stiffness (kPa) | | 4.5 | (3.6;5.6) | 4.9 | (4.2;5.7) | 4.2 | (3.5;5.5) | **0.018** | |
| Low risk | < 8 kPa | 114 | (95%) | 47 | (94%) | 67 | (95.7%) | 0.67 | |
| Increased risk | ≥ 8 kPa | 6 | (5%) | 3 | (6%) | 3 | (4.3%) | | |
| NFS[a] (n = 117) | | -1.4 | (-2.13;-0.21) | -1.15 | (-1.81;-0.20) | -1.62 | (-2.41;-0.49) | 0.15 | |
| | ≥ Sens. Cut-off | 35 | (29.9%) | 15 | (31.3%) | 20 | (29.0%) | 0.79 | |
| | ≥ Spec. Cut-off | 9 | (7.7%) | 5 | (10.4%) | 4 | (5.8%) | 0.36 | |
| FIB4[b] (n = 118) | | 1.5 | (1.12;1.94) | 1.64 | (1.26;2.16) | 1.47 | (1.03;1.84) | 0.064 | |
| | ≥ Sens. Cut-off | 53 | (44.9%) | 25 | (51.0%) | 28 | (40.6%) | 0.26 | |
| | ≥ Spec. Cut-off | 10 | (8.5%) | 7 | (14.3%) | 3 | (4.3%) | 0.056 | |
| **Indicators of steatohepatitis** | | | | | | | | | |
| AST % of ULN | | 60 | (50.6;76.7) | 60 | (50.6;89.6) | 60 | (51.7;69.9) | 0.35 | |
| | AST > ULN **(n)** | 12 | (10%) | 9 | (18%) | 3 | (4.3%) | **0.014** | |
| ALT % of ULN | | 57.3 | (45.6;75.9) | 64.8 | (46.2;82.1) | 55.3 | (45.7;70.7) | 0.20 | |
| | ALT > ULN **(n)** | 11 | (9.2%) | 10 | (20%) | 1 | (1.4%) | **<0.001** | |
| FAST[c] | | 0.15 | (0.10;0.25) | 0.22 | (0.13;0.34) | 0.12 | (0.09;0.21) | **<0.001** | |
| | ≥ Sens. Cut-off | 16 | (13.3%) | 12 | (24%) | 4 | (5.7%) | **0.004** | |
| | ≥ Spec. Cut-off | 3 | (2.5%) | 3 | (6%) | 0 | (0%) | | |

Values given in median (IQR) and absolute numbers (%)

CAP, Controlled Attenuation Parameter; NFS, NAFLD-Fibrosis Score; FIB4, FIB4-index; AST, aspartate aminotransferase; ALT, alanine aminotransferase; ULN, upper limit of normal; FAST, Fibrosis-AST-score

[a] sensitive/specific cut-offs were −1.455 (age 36–65) and 0.12 (age ≥ 65)/0.676 (age ≥ 36)

[b] sensitive/specific cut-offs were 1.3 (age < 65) and 2.0 (age ≥ 65)/2.67 (all ages)

[c] sensitive/specific cut-offs were 0.35/0.67

Our cohort represents the typical spectrum of patients at risk of coronary artery disease with an expectable rate of coronary intervention [43, 44]. 41% of patients fulfilled the MAFLD definition, which is slightly higher compared to the estimated NAFLD prevalence in the

**Table 3. Variables associated with coronary intervention using multivariate logistic regression analysis.**

| | Estimate (OR) | 95% CI | p-value |
|---|---|---|---|
| **Sex (female vs male)** | 0.25 | 0.09 to 0.66 | 0.007 |
| **Age (per 10 years)** | 1.66 | 1.05 to 2.76 | 0.038 |
| **BMI (per kg/m²)** | 0.90 | 0.81 to 1.00 | 0.054 |
| **Smoking (yes vs no)** | 1.25 | 0.49 to 3.29 | 0.64 |
| **Risky alcohol consumption (yes vs no)[a]** | 0.80 | 0.32 to 1.96 | 0.63 |
| **FAST score (per log odds ratio)** | 2.28 | 1.40 to 3.96 | 0.001 |

OR, Odds ratio; CI, confidence interval; BMI, Body-Mass index; FAST, Fibrosis-AST score

[a] >20g/day in men and >10g/day in women

general adult population in Germany [20]. However, we only observed a low frequency (5%) of cases at risk of advanced liver disease defined by elevated VCTE values or serum-based fibrosis markers in the total cohort, which converts to a prevalence of advanced liver disease of approximately 15% in the MALFD sub-group. This is in line with a recent observational study from Germany that revealed an overall prevalence of relevant fibrosis of 19% among NAFLD patients at secondary referral institutions [45]. Interestingly, the traditional risk factors diabetes, obesity and arterial hypertension were neither strongly associated with advanced liver disease nor with the need for cardiovascular revascularization therapy. This might result from the relatively small study population and may reflect the small effect size in preselected cases. Only male gender had a significant association with CAD risk which is in line with reported data and due to the practice of patient selection at the time of recruitment, in which women show a higher rate of false positive ergometry and myocardial perfusion scintigraphy [44, 46].

Our findings demonstrate, that FLD prevalence in CAD patients shows no relevant deviation from age-adjusted values from the normal population. Hence, a close interplay between the drivers of CAD and FLD progression remains questionably, and thus, intensified screening measures for advanced FLD in patients undergoing CA beyond the actual recommendations [18, 20] seems not indicated for routine medical care. In addition, the poor correlation between different non-invasive markers of advanced fibrosis in our cohort underlines that confounding factors, e.g., medication or intervention interfering with platelet counts, must be taken into account when assessing FLD risk. However, patients at risk for advanced liver disease (e.g. with metabolic syndrome) should undergo screening as recommended by the current guideline independent of the presence CAD [20, 47].

As a secondary finding, we observed associations of higher liver stiffness within the limits of normal in patients with relevant CAD compared to those without need of revascularization therapy. Even after exclusion of patients with LSM ≥8 kPa, the difference remained significant (4.8 kPa (IQR 4.1;5.5) vs. 4.1 (IQR 3.3;5.2) kPa; p = 0.018). Elevated liver stiffness at lower levels does not necessarily reflect tissue fibrosis but is also an indicator of tissue vascularization and inflammation [25, 48]. These observations were incorporated in the development of the FAST score, which reflects the risk of fibrotic steatohepatitis with good accuracy [27] and has meanwhile gained attention as potential guidance in the referral pathways of patients with FLD [31, 49]. Higher FAST score values were the strongest predictor of relevant CAD in our cohort. This phenomenon has not been reported before and points to inflammatory-driven links between CAD and FLD progression. Low grade systemic inflammation has been described in many metabolic diseases including Type-2-Diabetes-mellitus, obesity and NAFLD [40] and is also relevant in cardiovascular pathologies [50]. Therefore, future studies should focus on the potential role of FAST score as a stratification method not only in the field of liver diseases but also in settings where cardiac risk is the dominant question. Ideally, this could be accompanied by analysis of liver tissue, visceral fat and endothelial function to get further insights in the mutual pathophysiological mechanisms of these disease.

Our study has several limitations. The monocentric design restricted the case numbers and may have led to selection bias according to local healthcare pathways. However, our cohort is comparable to other studies comparing CAD risk and liver disease [51, 52]. In this pilot study, we chose a dichotomous endpoint (presence of CAD requiring intervention). Further correlation with the severity of CAD would have required pressure analysis for estimation of coronary flow reserve and microvascular resistance index [19], which was not always available in our patients. Moreover, we could only use non-invasive indicators of FLD severity instead of liver histology. This reference standard is, however, associated with relevant bleeding risk in cohorts with the need of anticoagulation therapy, which is very common among cardiovascular patients. Although there is no general recommended cut-off for CAP, we used the cut-offs

proposed by Eddowes et al. [34], one of the largest prospective biopsy-controlled studies providing data in CAP accuracy for steatosis quantification. Established non-invasive markers are therefore an accepted substitute for liver biopsy [20, 48] and may help to avoid selection bias.

## Conclusion

In conclusion, our data show that the prevalence of advanced FLD is low in CAD patients requiring invasive procedures. Noninvasive estimates of fibrosis and steatosis severity were not associated with the need for coronary intervention. Elevated FAST score values underline the pathophysiological importance of inflammatory activity in both FLD and CAD.

## Supporting information

**S1 Table. Severity of liver disease on the subgroup of patients with MAFLD.**
(DOCX)

**S2 Table. Characterization of patients at high and low risk of relevant liver fibrosis.**
(DOCX)

**S1 Dataset.**
(XLSX)

## Acknowledgments

We thank Dr. David Petroff, University of Leipzig, Clinical Trial Centre Leipzig, for his advice and assistance for the statistical analyses.

## Author Contributions

**Conceptualization:** Sebastian Beer, Neef Martin, Johannes Wiegand, Thomas Karlas.

**Data curation:** Sebastian Beer, Jonas Babel, Thomas Karlas.

**Formal analysis:** Sebastian Beer, Jonas Babel, Thomas Karlas.

**Funding acquisition:** Sebastian Beer.

**Investigation:** Sebastian Beer, Jonas Babel.

**Methodology:** Sebastian Beer, Neef Martin, Thomas Karlas.

**Project administration:** Thomas Karlas.

**Resources:** Thomas Karlas.

**Supervision:** Sebastian Beer, Thomas Karlas.

**Validation:** Sebastian Beer, Jonas Babel, Thomas Karlas.

**Visualization:** Jonas Babel, Thomas Karlas.

**Writing – original draft:** Sebastian Beer, Jonas Babel, Thomas Karlas.

**Writing – review & editing:** Sebastian Beer, Jonas Babel, Neef Martin, Valentin Blank, Johannes Wiegand, Thomas Karlas.

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
