## [Decision Letter · Decision Letter 0]

6 Mar 2023

PONE-D-23-00204Non-invasive assessment of steatohepatitis indicates increased risk of coronary artery diseasePLOS ONE

Dear Dr. Beer,

Thank you for submitting your manuscript to PLOS ONE. After careful consideration, we feel that it has merit but does not fully meet PLOS ONE’s publication criteria as it currently stands. Therefore, we invite you to submit a revised version of the manuscript that addresses the points raised during the review process.

 Please submit your revised manuscript by Apr 20 2023 11:59PM. If you will need more time than this to complete your revisions, please reply to this message or contact the journal office at plosone@plos.org. Please include the following items when submitting your revised manuscript:A rebuttal letter that responds to each point raised by the academic editor and reviewer(s). You should upload this letter as a separate file labeled 'Response to Reviewers'.A marked-up copy of your manuscript that highlights changes made to the original version. You should upload this as a separate file labeled 'Revised Manuscript with Track Changes'.An unmarked version of your revised paper without tracked changes. You should upload this as a separate file labeled 'Manuscript'.

We look forward to receiving your revised manuscript.

Kind regards,

Shunsuke Mori, MD, PhD

Academic Editor

PLOS ONE

Journal Requirements:

2. "Thank you for stating the following in the Competing Interests section: 

    "I have read the journal's policy and the authors of this manuscript have the following competing interests: TK and JW received an unrestricted research grant from Echosens, France. TK was an invited speaker for Echosens.SB, JB, MN and VB declared that no competing interests exist"

Additional Editor Comments:

Our reviewers found some merits in this article but made a number of comments and suggestions. I ask the authors to clarify/correct each of the points that the reviewers indicated below. Additionally, in the text and tables, the authors should report p values with up to 2 significant digits and maximum 3 decimals (p = 0.21, p = 0.007, p = 0.001, etc.).

Reviewers' comments:

Reviewer's Responses to Questions

**Comments to the Author**

1. Is the manuscript technically sound, and do the data support the conclusions?

Reviewer #1: Yes

Reviewer #2: Yes

2. Has the statistical analysis been performed appropriately and rigorously? 

Reviewer #1: Yes

Reviewer #2: Yes

3. Have the authors made all data underlying the findings in their manuscript fully available?

Reviewer #1: No

Reviewer #2: Yes

4. Is the manuscript presented in an intelligible fashion and written in standard English?

Reviewer #1: Yes

Reviewer #2: Yes

5. Review Comments to the Author

Reviewer #1: Reviewer : The authors have prospectively evaluated non-invasive fibrosis tests (FIB4, NFS, FAST and Fibroscan) in 120 patients with coronary arterial disease. MAFLD and advanced liver fibrosis (LSM ≥8 kPa) were observed in 41% and 6% (5% in the abstract?), respectively. A significant high score was present in patients needing coronary intervention.

Paragraph “Introduction”

• Page 3, line 73-74: The authors could add the more recent reference by Toh MR, (Global epidemiology and genetics of hepatocellular carcinoma. Gastroenterology 2023 Feb 2:S0016-5085(23)00105-1.) Tho et al. indicated that “Non-cirrhotic HCC is more common in NASH than the other etiologies; 39% of NASH-associated HCCs develop in absence of cirrhosis, compared with 22%, 6%, 9% for HBV, HCV, ALD respectively.”

• Page 3, line 78: Please, indicate the precise prevalence with its CI : 44.6% (95% CI, 36.0%–53.6%). Interestingly, the prevalence of CAD was even more important when invasive coronary angiogram was used : 60.5% (95% CI, 43.8%–75.1%)

Paragraph “Materials and Methods”

• Page 4, line 128: The authors need to specify conditions of blood samples. Surely, in a fasting condition

• Page 5: The authors could add the reference by Castera L, et al (Gastroenterology 2019) for the threshold (8 kPa) used to define significant fibrosis

• Page 5, line 147 : “Patients with SLCD exceeding 25 mm were examined with the XL-probe”. This recommendation is similar to that also proposed by EASL using a BMI cut-off of 30 kg/m2 to select the M or XL probe (EASL-ALEH clinical practice guidelines. J Hepatol 2015;63:237–264). The authors could add this BMI cut-off for clinicians since BMI cut-off is more often used than SLCD.

• Page 5, line152: The authors used AAP value ≥302 dB/m to diagnose any grade of steatosis. However, different cut-off values have been proposed elsewhere and, for EASL, a CAP (> 275 dB/m) is currently suggested (J Hepatol. 2021; 75:659-689). This point could be discussed.

Paragraph “Results”

• In this section, baseline biological and demographic characteristics of the 120 patients should be clearly reported in a dedicated table. All the components of the MAFLD definition need to be reported (waist circumference, hs-CRP, etc.) together with liver enzymes, PT, albumin, platelets, etc. and % of patients with MAFLD.

• Instead of describing characteristics of patients with and without coronary intervention in the total cohort, the authors should complete this table 1 by describing the non-MAFLD population

• In this section, the authors could specify the indications of routine (n= …) or emergency angiography (n=…).

• Page 9 : the authors write “LSM was valid and reliable in 120 (94% of eligible) patients”. The 128 “eligible” patients could be shown in the flow chart (Figure 1) for a more comprehensive reading.

• Table 2 : the authors need to define the ULN of AST and ALT in their centre. All percentages have been rounded in this table. I suggest to indicate the right numbers (for instance 4/70 = 5.7% instead of 6%). The authors may round figures in the discussion

• Page 11, line 241 : the authors write “None of the high-risk FIB4 patients and only one of the high-risk NFS patients had increased LSM values”. I suggest to create a table in supplementary appendix to characterize the 113 patients with a low risk (< 8kPa) and the 7 patients with a high risk (≥ 8 kPa), even though the number of high-risk cases is low. We need to better characterize the small group of CAD patients with LSM ≥ 8 kPa

• Table 3 : the title of this table does not indicate clearly what the authors are really looking for. I suggest to writ “Variables associated with coronary intervention using multivariate analysis”.

• Page 11, line 249 : The authors write “The results are very similar in the MAFLD subgroup”. The authors provide in Suppl. Appendix regarding this multivariate analysis regarding the subgroup of 49 MAFLD patients. Since only 22 coronary interventions were performed in this subgroup, only two variables can be entered in the model.

Paragraph “Discussion”

• Page 12, line 273-74: It is surprising that metabolic factors are not associated with advanced liver fibrosis in this study. Indeed, liver fibrosis development in NAFLD patients may result from a long-term exposure to cardiometabolic risk factors such as diabetes, arterial hypertension, dyslipidaemia and overweight (Baratta F, et al. Gastroenterol Hepatol 2020;18:2324-31). Genetic factors may also impact the relationship between NAFLD and CAD (Miao Z, et al. HGG Adv 2021 Aug 24;3:100056) ; The pathophysiology of the relationship between NAFLD and CAD could be explained briefly with the main causal factors.

• Page 12, line 285: “As a secondary finding, we observed associations of higher liver stiffness within the limits of normal in patients with relevant CAD compared to those without need of revascularization therapy” : these data are not shown.

• Please, check the references 7, 8, 9, etc.

Reviewer #2: The study presents the results of an interesting and original research. Experiments, statistics, and other analyses are performed to a high technical standard and are described in a detailed manner.

Conclusions are presented in an appropriate fashion and are supported by the data, but the authors must describe if are any group of patients with coronary interventions that must do the non-invasive assessment of liver fibrosis, probably patients with metabolic syndrome or T2DM should be examined with transient elastography for liver fibrosis. Also, the authors must explain if are any correlations between LSM values and severity of CAD according to the grade of stenosis. Also if are any correlations between them in which patients are most commen met?

In addition, they must put a space before the references in thes text.

6. PLOS authors have the option to publish the peer review history of their article (what does this mean?). If published, this will include your full peer review and any attached files.

Reviewer #1: **Yes: **Thevenot

Reviewer #2: No

---

## [Author Response · Author response to Decision Letter 0]

18 Apr 2023

All comments of the reviewers are addressed in the file "Response to the reviewers", identical to the following: 

EDITORS' COMMENTS TO THE AUTHOR:

We thank the editor for this opportunity. We fully revised the manuscript according to all style requirements of PLOSone including the file naming. We hope we did everything fully sufficiently. 

2. "Thank you for stating the following in the Competing Interests section: 

 "I have read the journal's policy and the authors of this manuscript have the following competing interests: TK and JW received an unrestricted research grant from Echosens, France. TK was an invited speaker for Echosens.SB, JB, MN and VB declared that no competing interests exist"

We updated the statement for competing interests accordingly. 

We ensured that the ORCID iD of the corresponding author validated and updated. 

We listed the Supporting Information accordingly at the end of the manuscript and updated the in-text citation.

Additional Editor Comments:

Our reviewers found some merits in this article but made a number of comments and suggestions. I ask the authors to clarify/correct each of the points that the reviewers indicated below. Additionally, in the text and tables, the authors should report p values with up to 2 significant digits and maximum 3 decimals (p = 0.21, p = 0.007, p = 0.001, etc.).

The p values have been adapted. 

PPOINT-TO-POINT RESPONSE TO REVIEWER'S COMMENTS TO THE AUTHOR

REVIEWER #1

Have the authors made all data underlying the findings in their manuscript fully available?

We thank the reviewer for the comment. We updated the supporting file labeling and listed the files at the end of the manuscript. The complete raw data used for this manuscript is included in S1 Dataset. 

The authors have prospectively evaluated non-invasive fibrosis tests (FIB4, NFS, FAST and Fibroscan) in 120 patients with coronary arterial disease. MAFLD and advanced liver fibrosis (LSM ≥8 kPa) were observed in 41% and 6% (5% in the abstract?), respectively. A significant high score was present in patients needing coronary intervention.

We thank the reviewer for note and corrected the discrepancy in the percentage. 

Paragraph “Introduction”

• Page 3, line 73-74: The authors could add the more recent reference by Toh MR, (Global epidemiology and genetics of hepatocellular carcinoma. Gastroenterology 2023 Feb 2:S0016-5085(23)00105-1.) Tho et al. indicated that “Non-cirrhotic HCC is more common in NASH than the other etiologies; 39% of NASH-associated HCCs develop in absence of cirrhosis, compared with 22%, 6%, 9% for HBV, HCV, ALD respectively.”

We are thankful for the advice in additional references and amended accordingly.

• Page 3, line 78: Please, indicate the precise prevalence with its CI : 44.6% (95% CI, 36.0%–53.6%). Interestingly, the prevalence of CAD was even more important when invasive coronary angiogram was used: 60.5% (95% CI, 43.8%–75.1%)

We provided that important additional information fittingly. 

Paragraph “Materials and Methods”

• Page 4, line 128: The authors need to specify conditions of blood samples. Surely, in a fasting condition

The missing statement “in an overnight fasting state” was added. 

• Page 5: The authors could add the reference by Castera L, et al (Gastroenterology 2019) for the threshold (8 kPa) used to define significant fibrosis

The additional reference was included. 

• Page 5, line 147 : “Patients with SLCD exceeding 25 mm were examined with the XL-probe”. This recommendation is similar to that also proposed by EASL using a BMI cut-off of 30 kg/m2 to select the M or XL probe (EASL-ALEH clinical practice guidelines. J Hepatol 2015;63:237–264). The authors could add this BMI cut-off for clinicians since BMI cut-off is more often used than SLCD.

We thank the reviewer for this suggestion. For routine care BMI might also be used to determine the choice of M vs. XL probe. However, we think that the choice according to SLD is more accurate. The recent version of the Fibroscan device is equipped with an automated-probe selection tool that is based on SLD assessment using the A-mode. Therefore, we used a standardized probe choice approach that has been described in Blank et al. (Dig Liver Dis. 2022 Mar;54(3):358-364. doi: 10.1016/j.dld.2021.08.003.) This paper also discusses the issue of a BMI based probe choice in more detail. We have added a reference to this paper.

• Page 5, line152: The authors used AAP value ≥302 dB/m to diagnose any grade of steatosis. However, different cut-off values have been proposed elsewhere and, for EASL, a CAP (> 275 dB/m) is currently suggested (J Hepatol. 2021; 75:659-689). This point could be discussed.

Recent analysis on the accuracy of CAP show that the specificity of this technology can be impaired by BMI and patients’ anthropometry, especially in NAFLD patients [Petroff, Blank et al. Lancet Gastroenterol Hepatol. 2021 Mar;6(3):185-198. doi: 10.1016/S2468-1253(20)30357-5.]. Therefore, the choice of the cut-off depends on the clinical strategy (“ruling in” vs. “ruling out”) and the local prevalence. The applied cut-off derives from the largest prospective trial on NAFLD screening using VCTE/CAP including > 400 biopsy-controlled data sets from 7 centers in the UK [Eddowes et al. Gastroenterology. 2019 May;156(6):1717-1730. doi: 10.1053/j.gastro.2019.01.042]. The screening approach and patients’ characteristics of our study has many parallels to that of Eddowes at al. Therefore, we refrain from introducing other cut-off values.

Therefore, we added in the discussion: 

“Although there is no general recommended cut-off for CAP, we used the cut-offs proposed by Eddowes et al. [Ref], one of the largest prospective biopsy-controlled studies providing data in CAP accuracy for steatosis quantification.”

Paragraph “Results”

• In this section, baseline biological and demographic characteristics of the 120 patients should be clearly reported in a dedicated table. All the components of the MAFLD definition need to be reported (waist circumference, hs-CRP, etc.) together with liver enzymes, PT, albumin, platelets, etc. and % of patients with MAFLD.

We modified Table 1 accordingly by extending the reported values.

• Instead of describing characteristics of patients with and without coronary intervention in the total cohort, the authors should complete this table 1 by describing the non-MAFLD population

Modification of table 1 included a comparison of MAFLD and non-MAFLD patients. 

• In this section, the authors could specify the indications of routine (n= …) or emergency angiography (n=…).

Number and indication of angiography were specified: “Of these 120 patients, n=19 (15.8%) underwent emergency CA due to (suspected) myocardial infarction. Indication for routine CA (n=101) was stable angina pectoris.”

• Page 9 : the authors write “LSM was valid and reliable in 120 (94% of eligible) patients”. The 128 “eligible” patients could be shown in the flow chart (Figure 1) for a more comprehensive reading.

Figure 1 was adjusted correspondingly. 

• Table 2 : the authors need to define the ULN of AST and ALT in their centre. All percentages have been rounded in this table. I suggest to indicate the right numbers (for instance 4/70 = 5.7% instead of 6%). The authors may round figures in the discussion

The ULN have been added to the laboratory methods: “The upper limit of normal of AST at time of the study was 0.6 µkat/l in women and 0.85 µkat/l in men, and of ALT 0.58 µkat/l in women and 0.85 µkat/l in men.” The format of numbers was adapted. 

• Page 11, line 241 : the authors write “None of the high-risk FIB4 patients and only one of the high-risk NFS patients had increased LSM values”. I suggest to create a table in supplementary appendix to characterize the 113 patients with a low risk (< 8kPa) and the 7 patients with a high risk (≥ 8 kPa), even though the number of high-risk cases is low. We need to better characterize the small group of CAD patients with LSM ≥ 8 kPa

We presented an additional S2 Table for a more detailed characterization of the patients with significantly elevated LSM and added to the results: “The small number of patients with significant elevated LSM ≥8kPa had significantly higher BMI, waist-to-hip ratio and CAP resulting in an increased prevalence of MAFLD and elevated FAST score (S2 Table).” 

• Table 3 : the title of this table does not indicate clearly what the authors are really looking for. I suggest to write “Variables associated with coronary intervention using multivariate analysis”.

We modified the title according to the reviewer’s recommendation. 

• Page 11, line 249 : The authors write “The results are very similar in the MAFLD subgroup”. The authors provide in Suppl. Appendix regarding this multivariate analysis regarding the subgroup of 49 MAFLD patients. Since only 22 coronary interventions were performed in this subgroup, only two variables can be entered in the model.

We thank the reviewer for this critical remark. We have adjusted the respective phrase accordingly: “In the MAFLD subgroup the values were similar although the limited patient number did not allow a multivariate logistic regression.”

Paragraph “Discussion”

• Page 12, line 273-74: It is surprising that metabolic factors are not associated with advanced liver fibrosis in this study. Indeed, liver fibrosis development in NAFLD patients may result from a long-term exposure to cardiometabolic risk factors such as diabetes, arterial hypertension, dyslipidaemia and overweight (Baratta F, et al. Gastroenterol Hepatol 2020;18:2324-31). Genetic factors may also impact the relationship between NAFLD and CAD (Miao Z, et al. HGG Adv 2021 Aug 24;3:100056) ; The pathophysiology of the relationship between NAFLD and CAD could be explained briefly with the main causal factors.

We thank the reviewer for this comment. We added the following in the discussion: “This might result from the relatively small study population and may also reflect the small effect size in preselected cases.” Further pathophysiological factors such as genetic alterations are involved in the relationship between NAFLD and CAD [(Miao Z, et al. HGG Adv 2021 Aug 24;3:100056] but could not be addressed in our study.

• Page 12, line 285: “As a secondary finding, we observed associations of higher liver stiffness within the limits of normal in patients with relevant CAD compared to those without need of revascularization therapy”: these data are not shown.

We endorsed our statement by calculation and the following additional statement: “Even after exclusion of the patients with LSM ≥8 kPa, difference remained significant (4.8 kPa vs. 4.1 kPa; p=0.018)”

• Please, check the references 7, 8, 9, etc.

The missing details have been added. 

Reviewer #2: 

The study presents the results of an interesting and original research. Experiments, statistics, and other analyses are performed to a high technical standard and are described in a detailed manner.

Conclusions are presented in an appropriate fashion and are supported by the data, but the authors must describe if are any group of patients with coronary interventions that must do the non-invasive assessment of liver fibrosis, probably patients with metabolic syndrome or T2DM should be examined with transient elastography for liver fibrosis. 

We thank the reviewer for this critical remark and have adjusted the discussion accordingly:

“However, patients at risk for advanced liver disease (e.g. with metabolic syndrome) should undergo screening as recommended by current guidelines independent of the presence CAD. [Tacke F, et al.. Updated S2k Clinical Practice Guidelines: NAFLD. Z Gastroenterol. 2022; 60(9):e733-e801.; Visseren FLJ, et al. 2021 ESC Guidelines on cardiovascular disease prevention in clinical practice. Eur Heart J. 2021; 42(34):3227–337.]” 

Also, the authors must explain if are any correlations between LSM values and severity of CAD according to the grade of stenosis. Also if are any correlations between them in which patients are most commen met?

We thank the reviewer for this suggestion. We had discussed this idea during the preparation of the study protocol, but finally chose to define a dichotomous endpoint (CAD present or not), because any further correlation with disease severity would go beyond the scope of a pilot study of this scale. Especially, the severity of CAD cannot be simply correlated with a single factor (such as the grade of stenosis), but would require more special methods including pressure analysis for estimation of coronary flow reserve and microvascular resistance index (Knuuti J, et al.: 2019 ESC Guidelines for the diagnosis and management of chronic coronary syndromes: The Task Force for the diagnosis and management of chronic coronary syndromes of the European Society of Cardiology (ESC). European Heart Journal 2020; 41 (3): 407–7).

We added a phrase to the discussion.

In addition, they must put a space before the references in the text.

This was corrected.

---

## [Decision Letter · Decision Letter 1]

4 May 2023

PONE-D-23-00204R1Non-invasive assessment of steatohepatitis indicates increased risk of coronary artery diseasePLOS ONE

Dear Dr. Karlas,

Thank you for submitting your manuscript to PLOS ONE. After careful consideration, we feel that it has merit but does not fully meet PLOS ONE’s publication criteria as it currently stands. Therefore, we invite you to submit a revised version of the manuscript that addresses the points raised during the review process.

This second version of the manuscript has been revised well. Reviewer #1 requested the authors to clarify just a few points. Please reply to the reviewer’s comments and make appropriate revision if necessary.

We look forward to receiving your revised manuscript.

Kind regards,

Shunsuke Mori, MD, PhD

Academic Editor

PLOS ONE

Journal Requirements:

Reviewers' comments:

Reviewer's Responses to Questions

**Comments to the Author**

1. If the authors have adequately addressed your comments raised in a previous round of review and you feel that this manuscript is now acceptable for publication, you may indicate that here to bypass the “Comments to the Author” section, enter your conflict of interest statement in the “Confidential to Editor” section, and submit your "Accept" recommendation.

Reviewer #1: All comments have been addressed

Reviewer #2: All comments have been addressed

2. Is the manuscript technically sound, and do the data support the conclusions?

Reviewer #1: Yes

Reviewer #2: Yes

3. Has the statistical analysis been performed appropriately and rigorously? 

Reviewer #1: Yes

Reviewer #2: Yes

4. Have the authors made all data underlying the findings in their manuscript fully available?

Reviewer #1: Yes

Reviewer #2: Yes

5. Is the manuscript presented in an intelligible fashion and written in standard English?

Reviewer #1: Yes

Reviewer #2: Yes

6. Review Comments to the Author

Reviewer #1: Minor corrections to provide :

Page 4, line 115-117: the references should be put at the end of the sentence : “….severe congestive heart failure …………… and pulmonary hypertension (WHO III or IV) (28, 29).”.

Revised Table 1 : Quantitative data should be expressed in median with IQR and qualitative data in absolute numbers with %

For instance :

“Male gender n (%): 80 (67%)” and only one column is sufficient

The line “AST % of ULN” is write twice ?

AST should be expressed in µkat/L, like GGT and AP

Please correct “gGT” by “GGT”

Instead of “exp9/l”, write “109/L” or “mm-3”

Is is really useful to write “HbA1c NSGP/DCCT”? It should be simpler to write only “HbA1c”

Page 9, line 238: write “≥ S2”

Page 11 : In the following sentence, the authors could specify that “high-risk”patients were defined according to the specific cut-off of NFS : “Only 5 patients (10%) of the interventional group and 4 patients (6%) of the non- intervention group (p=0.36) were identified as high-risk”

Page 11, line 266 : it would be better to write in full “MI” for a better understanding of this sentence

Reviewer #2: The work of Beer et al. is an exceptional original paper, which includes new data regarding liver fibrosis and CAD. Also, this paper open a new path in the links between liver fibrosis and CAD.

7. PLOS authors have the option to publish the peer review history of their article (what does this mean?). If published, this will include your full peer review and any attached files.

Reviewer #1: **Yes: **Thevenot

Reviewer #2: No

---

## [Author Response · Author response to Decision Letter 1]

21 May 2023

REVIEWER #1

Page 4, line 115-117: the references should be put at the end of the sentence: “….severe congestive heart failure …………… and pulmonary hypertension (WHO III or IV) (28, 29).”.

We corrected this accordingly.

Revised Table 1 : Quantitative data should be expressed in median with IQR and qualitative data in absolute numbers with %

For instance :

“Male gender n (%): 80 (67%)” and only one column is sufficient

The line “AST % of ULN” is write twice ?

AST should be expressed in µkat/L, like GGT and AP

Please correct “gGT” by “GGT”

Instead of “exp9/l”, write “109/L” or “mm-3”

Is is really useful to write “HbA1c NSGP/DCCT”? It should be simpler to write only “HbA1c”

The recommendations have been implemented. Formatting changes have been adapted also to table 2. 

Concerning AST and ALT values: We use sex-specific cut-off values for these parameters. Therefore, the absolute values (in µkat/L) do not provide sufficient information when presented alone. That is why we initially decided to show %-values in relation to the upper limit of normal (ULN) and provided an additional line for categorial numbers, i.e. cases with increased ALT or AST, respectively. We now added absolute AST and ALT values to Table 1, but we abstained to do so in Table 2 for ease of reading.

Page 9, line 238: write “≥ S2”

We corrected accordingly.

Page 11 : In the following sentence, the authors could specify that “high-risk”patients were defined according to the specific cut-off of NFS : “Only 5 patients (10%) of the interventional group and 4 patients (6%) of the non- intervention group (p=0.36) were identified as high-risk”

We added to the sentence: “according to the specific cut-off of NFS”

Page 11, line 266: it would be better to write in full “MI” for a better understanding of this sentence

We made the change accordingly. 

Reviewer #2: The work of Beer et al. is an exceptional original paper, which includes new data regarding liver fibrosis and CAD. Also, this paper open a new path in the links between liver fibrosis and CAD.

Thank you.

---

## [Editor Report · Decision Letter 2]

25 May 2023

Non-invasive assessment of steatohepatitis indicates increased risk of coronary artery disease

PONE-D-23-00204R2

Dear Dr. Karlas,

We’re pleased to inform you that your manuscript has been judged scientifically suitable for publication and will be formally accepted for publication once it meets all outstanding technical requirements.

Kind regards,

Shunsuke Mori, MD, PhD

Academic Editor

PLOS ONE

---

## [Editor Report · Acceptance letter]

5 Jun 2023

PONE-D-23-00204R2 

Non-invasive assessment of steatohepatitis indicates increased risk of coronary artery disease 

Dear Dr. Karlas:

I'm pleased to inform you that your manuscript has been deemed suitable for publication in PLOS ONE. Congratulations! Your manuscript is now with our production department. 

Kind regards, 

on behalf of

Dr. Shunsuke Mori 

Academic Editor

PLOS ONE